# Computer-Aided Diagnosis by Tissue Image Analysis as an Optical Biopsy in Hysteroscopy

**DOI:** 10.3390/ijms232112782

**Published:** 2022-10-24

**Authors:** Vasilios Tanos, Marios Neofytou, Panayiotis Tanos, Constantinos S. Pattichis, Marios S. Pattichis

**Affiliations:** 1Aretaeio Hospital, 55-57 Andrea Avraamides Street, Nicosia 2024, Cyprus; 2Medical School, Nicosia of University, 93 Agiou Nikolaou Street, Nicosia 2408, Cyprus; 3Department of Computer Science and Biomedical Engineering Research Center, University of Cyprus, Nicosia 1678, Cyprus; 4Causeway Hospital, Coleraine BT51 3AE, UK; 5Department of Electrical and Computer Engineering, The University of New Mexico, Albuquerque, NM 87131, USA

**Keywords:** hysteroscopy, eHealth, medical image processing, computer-aided diagnosis, tissue texture analysis, medical cameras, calibration, gamma correction

## Abstract

This review of our experience in computer-assisted tissue image analysis (CATIA) research shows that significant information can be extracted and used to diagnose and distinguish normal from abnormal endometrium. CATIA enabled the evaluation and differentiation between the benign and malignant endometrium during diagnostic hysteroscopy. The efficacy of texture analysis in the endometrium image during hysteroscopy was examined in 40 women, where 209 normal and 209 abnormal regions of interest (ROIs) were extracted. There was a significant difference between normal and abnormal endometrium for the statistical features (SF) features mean, variance, median, energy and entropy; for the spatial grey-level difference matrix (SGLDM) features contrast, correlation, variance, homogeneity and entropy; and for the gray-level difference statistics (GLDS) features homogeneity, contrast, energy, entropy and mean. We further evaluated 52 hysteroscopic images of 258 normal and 258 abnormal endometrium ROIs, and tissue diagnosis was verified by histopathology after biopsy. The YCrCb color system with SF, SGLDM and GLDS color texture features based on support vector machine (SVM) modeling correctly classified 81% of the cases with a sensitivity and a specificity of 78% and 81%, respectively, for normal and hyperplastic endometrium. New technical and computational advances may improve optical biopsy accuracy and assist in the precision of lesion excision during hysteroscopy. The exchange of knowledge, collaboration, identification of tasks and CATIA method selection strategy will further improve computer-aided diagnosis implementation in the daily practice of hysteroscopy.

## 1. Introduction

Gynecological cancer (ovarian, endometrial and cervical) consists of a large number of tumors with variable presentation and often unpredictable malignant potential, making patient management demanding and challenging. In addition, the necessity to preserve the patient’s sexuality and fertility potential and the requirement to offer optimum treatment to combat malignancies complicate management decisions even further. Surgery plays a significant role in the management of most gynecological benign and malignant tumors. Optimized first surgery may provide a survival advantage in this context. However, even the most experienced surgeon is totally reliant on intraoperative examination and histopathological evaluation for distinguishing between malignant and benign disease [1].

Repeated efforts in identifying high-risk gynecological cancer patients using imaging techniques (US, CT, MRI) and serum cancer markers did not improve the detection rate of early diagnosis in gynecological cancer. The naked eye can analyze color frequencies and detect shape and size differences of 100 microns in diameter. Minimally invasive surgery can facilitate and increase human sight limits up to 34 times that of the naked eye [2]. Hence, minimally invasive gynecological surgery is currently the gold standard procedure for the diagnosis and treatment of pelvic pathologies.

Hysteroscopy is replacing the traditional cervical dilatation and curettage for the diagnosis of endometrial cavity diseases, and laparoscopy has become the preferred surgical technique for treating benign and malignant gynecological conditions. Recent technological advances in telescope technology and small-size video cameras offering higher quality and higher resolution video acquisition helped to delineate new applications in the diagnosis and treatment of female reproductive organ pathology [3,4]. During minimally invasive surgery, the tissues are magnified and many pathological conditions can be seen that are very difficult to be identified by the naked eye. In this context, it is possible to diagnose cancer at an early stage; when linked with early initial treatment, this can save lives and limit the health complications and the suffering of patients, as well as minimize the surgery expenses.

Computer-assisted tissue characterization refers to the quantitative characterization of image features that lead to a highly probable distinction between normal and abnormal tissue. The appearance and structure of laparoscopy imaging make the use of texture analysis techniques suitable for computer-assisted tissue characterization purposes. The objective of this study was to characterize quantitatively the texture of gynecological tissue during endoscopy. This approach may lead to a highly probable distinction between normal and abnormal tissue in support of the doctor’s intraoperative diagnosis. A similar approach was also followed for colon endoscopic tissue characterization [5,6] for cancer from colonoscopic images.

In this article, we present our experience using CATIA for an optical biopsy in gynecological cases. An effort has been made to identify quantitative differences between normal and abnormal tissue using computer-aided diagnosis (CAD) as well as distinct tissue image characterization differences between benign and malignant tissue during minimally invasive gynecological surgery.

## 2. Results

### 2.1. Technical Conditions for Optimal Hysteroscopy Image Acquisition for CATIA

Hysteroscopy is considered the gold standard technique for the diagnosis and treatment of intrauterine pathologies and especially cancer [4]. A standardized protocol for capturing and analysis of hysteroscopy digital images for subsequent use in a CAD system for endometrial cancer has been developed based on the experiment presented in [7]. The abdominal cavity of a butchered chicken was used to compare texture feature variability under different viewing conditions in different angles (with 5 degrees difference) and different distances (at a distance of 3 cm in close-up view and 5 cm in panoramic view) from the imaged object [7]. The results indicated that for small consecutive angles, there is no significant difference in texture feature analysis, but there is a significant difference when comparing panoramic vs. close-up views [7]. In another study, our group demonstrated that some texture features can be used to differentiate between normal and abnormal endometrium images captured during hysteroscopy [8]. Image luminosity and focus normalization was addressed by the doctors by adjusting camera gain so that a medical band appears well balanced and in optimal focus on the screen. This approach is bound to introduce many sources of variability in the acquisition. Gamma color correction was used to standardize the image acquisition, irrespective of the optical system, and study its effect on both image classification and visual examination [7,8]. The gamma-corrected color images were visually better than the originals. There was no significant difference in texture features between the close-up and panoramic views or between angles, either before or after gamma correction. There was a significant difference in certain texture features between normal and abnormal endometrium, both before and after gamma correction. Based on these findings, a standardized protocol for the capturing and analysis of endoscopy digital images for subsequent use in a CAD system in endometrial cancer was proposed.

It is anticipated that the standardized capturing and analysis of endoscopy images will facilitate the creation of digital libraries of minimally invasive surgery images for a variety of tissues including endometrial, ovarian and possibly other organ cancers. Images could be captured in different operation camera settings during minimally invasive surgeries under different sessions from the same endoscopy imaging system (i.e., telescopes and light sources) and different endoscopy cameras and different endoscopy imaging systems. For capturing images in the latter two setups, further validation studies will be required.

### 2.2. Texture Analysis of the Endometrium during Hysteroscopy

The efficacy of texture analysis in the endometrium images during hysteroscopy was examined. Endometrium images were captured under three different conditions: (1) at optimum illumination and focus, images were frozen and digitized at 720 × 576 pixels using 24-bit color; (2) normal (*n* = 209) and abnormal (*n* = 209) regions of interest (ROIs) were manually selected by the physician; (3) images were captured using optimized viewing conditions, i.e., angle <5° and telescope tip to ROI distance <5 cm [7]. ROI images were converted into grayscale, and the statistical features (SF) and spatial gray-level dependence matrix (SGLDM) features and the gray-level difference statistics (GLDS) texture features were computed. Table 1 summarizes the median (P50%) and the 25th and 75th percentiles (P25% and P75%, respectively). The non-parametric Wilcoxon rank sum test at a = 0.05 was carried out to compare the differences between normal and abnormal tissue. As shown in Table 1, there was a significant difference between normal and abnormal endometrium for the SF features mean, variance, median, energy and entropy; for the SGLDM features contrast, correlation, variance, homogeneity and entropy; and for the GLDS features homogeneity, contrast, energy, entropy, and mean [7].

### 2.3. Assessing Endometrial Hyperplasia and Cancer

A prospective study evaluated 52 hysteroscopic images of 258 normal and 258 abnormal endometrial ROIs extracted manually by the gynecologist, and tissue diagnosis was verified by histopathology after biopsy [8]. The YCrCb color system with SF+GLDS color texture features based on support vector machine (SVM) modeling could correctly classify 81% of the cases with a sensitivity and specificity of 78% and 81%, respectively, for normal and hyperplastic endometrium. More specifically, the following features were statistically significantly different [8]: SF: *variance, entropy*; SGLDM: *ASM, contrast, variance, homogeneity, entropy*; GLDS: *homogeneity, contrast, energy, entropy, mean* for all color channels [8].

In another study by our group, using artificial neural networks, four texture features were selected to classify benign and malignant endometrium. The images were verified by the histopathologic diagnosis of 45 patients with normal endometrium and 10 patients with carcinoma. A classification accuracy of 89% and a specificity and sensitivity of 82% and 89%, respectively, were achieved.

## 3. Discussion

In practice, expert histopathologists with experience in each human body system cannot be found next to each surgeon. As a result, in most cases, therapy is based on postoperative histopathological diagnosis including second opinions, such as consultations from experts at remote sites. Ideally, however, patient benefit should be optimized at first surgery through timely and accurate histopathological diagnosis. Telepathology, CATIA and machine learning, including neural network classification, together can probably offer online dynamic intraoperative and postoperative consultations between a panel of experts via the transmission of video (laparoscopic and hysteroscopic scenes), still images (histopathologic specimen images) and clinical data. In this context, a digital library can be used by medical experts to interactively examine past similar case studies for preparatory research, follow-up research, archiving, training and standardization.

The main objective of this paper was to demonstrate the benefits of computer-assisted tissue image analysis to diagnose and distinguish normal from abnormal endometrium based on texture feature analysis and machine learning and neural network classification. The camera systems, monitors, operative techniques and skills developed with minimally invasive surgery provide tissue images and magnification with exceptional clarity. The abdomen and individual organs such as the endometrial cavity can be examined in situ with ease, without disturbing the anatomic features or the pathologic condition before treatment. Video images can be used intra- and postoperatively to re-evaluate the pathologic condition and provide the surgeon with excellent quality real-time video for assessing cavities and areas of the human body impossible to observe with the naked eye. The easy access to tissue images facilitates, encourages and accelerates the application of quantitative analysis using different algorithms, which are correlated with the histopathological findings [1,8].

Tissue visual signs, image texture analysis and features selected by machine learning and artificial neural network systems can serve as biomarkers for distinguishing abnormal from normal tissue. Precancerous as well as cancerous conditions are characterized as images with a complex set of attributes. Color, texture and relative geometry are predominately useful, while region shape is significantly less so. Regions are frequently amorphous or, for a few region classes, exhibit a shape that may be only approximately modeled, and even in these cases, the model may be image-dependent. The overall region of interest in the images may in general correlate with the histopathologic cancerous characteristics, such as abnormal tissue architecture, neo-angiogenesis, edema and cellular dysfunction. Images from a histopathologic section produced by microscopy may be interpreted by visual signs and tissue image features by computer-assisted diagnosis [9]. Such translation from microscopy tissue section characteristics to tissue image textures demands an allocation of data and computer system training [10]. CAD may have the potential to diagnose early disease, including cancer [1].

The loading of data with digital features of normal and abnormal tissue, with both visual and histopathologic characteristics, is essential in building the primary level of a CAD system. The functionality and efficiency of CAD depend on network capacity, speed of data processing and technological support [1]. Classical texture descriptors appear effective for texture characterization. Future work will include introducing different classification schemes [3]; augmenting the database, which is important in generalizing the results, especially when higher order statistics’ modeling is involved; and exploring the temporal dynamics of texture information, since taking information from neighbor frames may improve classification performance [11].

Experiments during hysteroscopy demonstrated that when three different texture feature algorithms, SF, SGDLM and GLDS, were used, CATIA results were reliable when the distance from the telescope tip to the tissue target was no more than 3 cm and the viewing angle was kept within 15 degrees deviation [7]. Selecting the best algorithm or combination of algorithms for the diagnosis of malignant tissue and new cases was a major challenge in almost all studies. CATIA technology needs to be adapted to clinical use, with real-time image analysis supported by a physician-friendly interface. The use of this technology for the diagnosis of malignancy is expected to diminish false negative results, a fact that is usually accompanied by an increase in false positives and a reduction in specificity.

In the studies we performed, the major advantage of CATIA was in comparing an abnormal tissue region to adjacent normal healthy tissue. Image comparisons can be performed during the intra- and postoperative periods to re-evaluate the pathologic features and operative technique. Easy access to tissue images facilitates, encourages and accelerates the application of quantitative image/video in hysteroscopy analysis by using different algorithms correlated with histopathologic findings [1,8].

## 4. Materials and Methods

The methodology components used to accomplish the computer-assisted tissue image analysis during minimally invasive gynecological surgery (MIGS) are presented below. Figure 1 presents a screenshot of a CAD system allowing the physician to crop normal and abnormal ROIs in the course of a hysteroscopy examination procedure. In addition, the corresponding ROI texture features are tabulated on the right side.

### 4.1. Standardized Protocol

The capturing protocol is demonstrated below in Figure 2, based on [8], supporting the minimum importing error for the images [12].

### 4.2. Recording of Endoscopic Video

For image acquisition, the medical telescopes provided by Wolf and STORZ have been used [2]. The telescope specifications were 2.8 mm diameter and 30° viewing angle for hysteroscopy and 10 mm diameter and 0° viewing angle for laparoscopy. Endoscopy video was captured using the Circon IP4.1 RGB video camera. All videos were captured at clinically optimum illumination and focusing. The camera was white-balanced using a white surface (white color of the palette [13]) as suggested by the manufacturer. The light source was a 300 W Xenon Light Source from ACMI Corporation [14]. The analog output signal from the camera (PAL 475 horizontal lines) was digitized at 720 × 576 pixels using 24-bit color and 25 frames per second at a resolution of approximately 15 pixels/mm for the panoramic view and approximately 21 pixels/mm for the close-up view. The video was saved in AVI format. Digitization was carried out using the Digital Video Creator 120 frame grabber that was connected to the PC through the IEEE 1394 port [15]. The capturing conditions were controlled by the physician, reflecting the clinical conditions of an operation. Moreover, the team used one more medical camera. A Storz three-chip camera, CO_2_ insufflator, cold light source and monitor were used for capturing more ROIs of the endometrium. In addition, illumination was adjusted for optimal viewing but not for calibrating results to include the viewing angle, distance and magnification of images. For both cases, live videos were recorded and analyzed by the CAD system.

“The Effect of Color Correction of Endoscopy Images for Quantitative Analysis in Endometrium” aimed to develop a standardized protocol for the capturing and analysis of hysteroscopic digital images for subsequent use in a computer-aided diagnosis (CAD) system in endometrial cancer. Hysteroscopic images were captured at optimum illumination and focus at 720 × 576 pixels using 24-bit color in the following cases: (i) for a variety of testing targets from a color palette with known color distribution, (ii) different viewing angles and distances from calf endometrium, and (iii) images from the human endometrium. Images were then gamma-corrected, and their classification performance was compared against that of non-gamma-corrected images. No significant difference in texture features was found between the close-up and panoramic views or between angles, either before or after gamma correction. There was a significant difference in certain texture features between normal and abnormal endometrium, both before and after gamma correction. These findings suggest that proper color correction can significantly impact CAD system performance, and its application prior to quantitative texture analysis in hysteroscopy is recommended.

### 4.3. Multiscale Texture Feature Variability Analysis of Images Captured under Different Viewing Positions

The variability of texture features for images of tissue captured under different viewing conditions was investigated by capturing the following sets of images: 20 images where the telescope tip was at a small distance from the tissue (close-up views, 3 cm distance for 10 images and 5 cm distance for 10 images) and 20 images for two consecutive angles (10 for each) with 5° difference [7]. Multiscale analysis was carried out in order to examine image texture at different scales. Images were downsampled and filtered to 10 scales (1 × 1 up to 10 × 10) for the different distances and 6 scales (1 × 1 up to 5 × 5 and 10 × 10) for the different angles. ROIs were selected from each image, and the following texture features were extracted: the SF and the SGLDM. Results indicate that there is significant variability between the panoramic and close-up views for multiscale texture features [7]. However, there is some variance (within reasonable bounds) between the multiscale texture features of consecutive angles [7]. The results of this experiment may prove useful in computer-aided diagnosis on images captured by hysteroscopy as well as in laparoscopy. Since the angles and tissue proximity from the hysteroscope tip are very small, seems that the CATIA error margin of hysteroscopic images is less and the reproducibility of the results is more robust. However, more experiments have to be carried out and more images have to be analyzed to support this further [16].

### 4.4. Texture Feature Extraction

ROIs were transformed into grayscale using the equation
Intensity =0.299×Red +0.587×Green +0.114×Blue channels
and the following texture features [17,18] were computed:

Statistical features (SF): SF features describe the gray-level histogram distribution without considering spatial independence. The following texture features were computed: (1) man, (2) variance and (3) entropy.

Spatial gray-level dependence matrix (SGLDM): The spatial gray-level dependence matrices as proposed by Haralick et al. [19] are based on the estimation of the second-order joint conditional probability density functions of two pixels (k, l) and (m, n) with distance d in the direction specified by the angle Θ having intensities of gray level i and gray level j. Based on the estimated probability density functions, the following 4 texture measures out of the 13 proposed by Haralick et al. [19] were computed: (1) contrast, (2) correlation, (3) homogeneity and (4) entropy. For a chosen distance d (in this work d = 1 was used) and for angles Θ = 0°, 45°, 90° and 135° we computed four values for each of the above texture measures. The above features were calculated for displacements δ = (0,1), (1,1), (1,0) and (1,−1), where = δ (Δx,Δy), and their ranges of values were computed.

Gray-level difference statistics (GLDS): The GLDS algorithm [20] is based on the assumption that useful texture information can be extracted using first-order statistics of an image. The algorithm is based on the estimation of the probability density p_δ_ of image pixel pairs at a given distance δ = (Δ_χ_, Δ_y_) having a certain absolute gray-level difference value. Let p_δ_ be the probability density of f_δ_ (x,y). If there are m gray levels, this has the form of an m-dimensional vector whose ith component is the probability that f_δ_ (x,y) will have the value i. If the picture f is discrete, it is easy to compute p_δ_ by counting the number of times each value of f_δ_ (x,y) occurs, where Δ_χ_ and Δ_y_ are integers. Coarse texture images result in low gray-level difference values, whereas fine texture images result in interpixel gray-level differences with great variances. Variable i is two pixels’ gray-level difference, m is the number of gray levels and pį is the individual probability. Features were estimated for the following distances: δ = (d,0), (d,d), (−d,d) and (0,d). A good way to analyze texture coarseness is to compute, for various magnitudes of į, some measure of the spread of values in p_δ_ away from the origin.

### 4.5. ROI Classification

The performance of the system using SVMs and probabilistic neural networks (PNNs) was also investigated. For both classifiers, training and testing for differentiating between normal ROIs and abnormal ROIs were performed. The C-SVM network was investigated using the Gaussian radial basis function (RBF) kernel and the linear kernel. Significantly better performance was obtained using the RBF kernel tuned based on the methodology proposed in [20]. More specifically, the values of c = 8 and γ = 0.04 were selected for prescribing the shape of the RBF kernel. These settings were fine-tuned based on numerous runs for different feature sets. We also consider the use of a PNN classifier that is based on the use of RBFs. This classifier was investigated for several spread radii in order to identify the optimal value following a similar procedure to that prescribed for the SVM case. The leave-one-out method was used for validating all the classification models unless otherwise stated. The runs were completed for each of the three color systems and for the Y channel. Better performance was obtained for the SVM models compared to the PNN classification models [8].

## 5. Conclusions

CAΤΙA needs to be further supported with well-designed studies enabling more extensive validation on larger datasets, in a clinical setting. When CATIA proves that it may increase the surgeon’s diagnostic ability and sampling precision, it could augment the intraoperative management decision and the surgeon’s performance. Additionally, it could minimize complications such as hemorrhage, hematoma, the spread of malignant cells, infection and scarring from multiple biopsies, as well as extensive tissue injuries. The proven efficacy of the discrimination ability of this CAD method, after validation by prospective and randomized studies, will allow the clinical implementation of CATIA systems linked with optical biopsies.

## Figures and Tables

**Figure 1 ijms-23-12782-f001:**
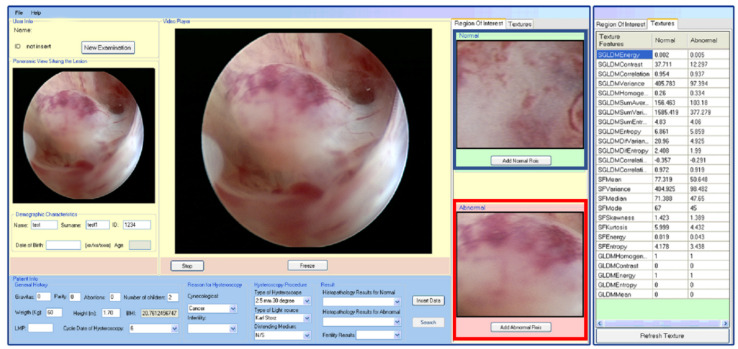
Illustration of a screenshot of a CAD system demonstrating video freezing and ROI selection and cropping, as well as the tabulation of the corresponding texture features. Blue-colored boundary: normal ROI. Red-colored boundary: abnormal ROI [8].

**Figure 2 ijms-23-12782-f002:**
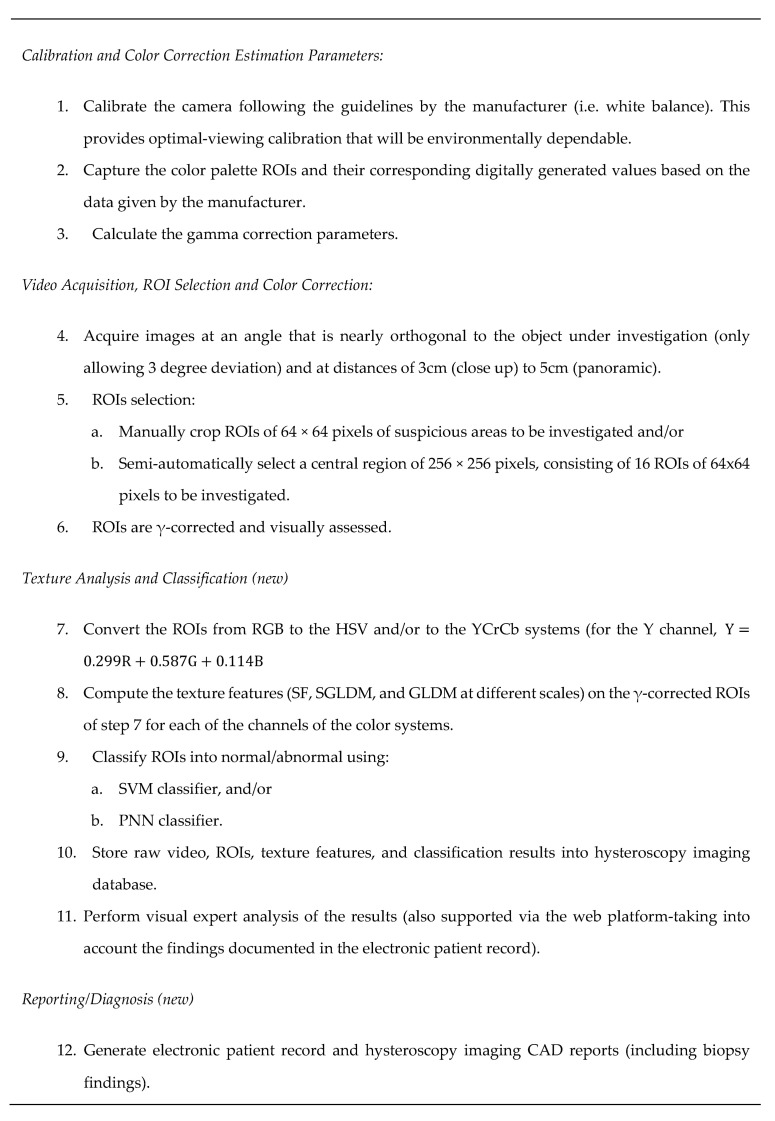
Hysteroscopy imaging CAD protocol covering image calibration and color correction, video acquisition, texture analysis and classification, and diagnosis. This protocol was updated from protocols previously published by our group in [8].

**Table 1 ijms-23-12782-t001:** Percentile values of the texture features and statistical analysis for normal (*n* = 209) vs. abnormal (*n* = 209) ROIs of the endometrium extracted from 40 women. Statistical analysis was carried out after gamma correction between the normal/abnormal ROIs at a ≤ 0.05 [7]. P25%, P50% and P75% correspond to the 25th, 50th (median) and 75th percentiles, respectively.

	Normal ROIs			Abnormal ROIs			Normal vs. Abnormal ROIs
	P25%	P50%	P75%	P25%	P50%	P75%	
SF							
Mean	138.44	156.06	173.91	129.37	144.65	170.48	1
Variance	29.44	54.63	127.94	66.9	124.39	223.33	1
Median	138.83	156.44	174.42	127.92	143.75	171.43	1
Energy	0.03	0.04	0.06	0.02	0.03	0.04	1
Entropy	3.02	3.34	3.68	3.44	3.74	3.99	1
SGLDM							
Contrast	3.1	3.82	4.87	4.82	7.04	10.99	1
Correlation	0.93	0.96	0.98	0.95	0.97	0.98	1
Variance	28.83	53.97	126.41	65.53	120.85	221.38	1
Homogeneity	0.45	0.48	0.5	0.38	0.42	0.46	1
Entropy	4.93	5.31	5.78	5.49	5.93	6.28	1
GLDS							
Homogeneity	0.45	0.48	0.5	0.38	0.42	0.46	1
Contrast	3.09	3.81	4.86	4.81	7.03	10.97	1
Energy	0.24	0.25	0.27	0.18	0.21	0.24	1
Entropy	1.45	1.54	1.64	1.63	1.77	1.96	1
Mean	1.33	1.44	1.63	1.62	1.89	2.31	1

## Data Availability

Not applicable.

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
