# Peer review of "Computer-Aided Diagnosis by Tissue Image Analysis as an Optical Biopsy in Hysteroscopy"

_ijms, 2022, doi:10.3390/ijms232112782_

Round 1

Reviewer 1 Report

Interesting and needed research on a timely topic. I have a few minor comments for the authors:
in the Abstract section: explain the abbreviation "ROI".  Swap the Discussion and Material and Methods sections in places, so that the methods are described first in the article. Unfortunately, the resolution and clarity of the figures presented are insufficient-try to improve this part of the paper-it will be of great benefit to the potential audience. Improve the References section-according to the guidelines for Authors in this journal.

Regards

Author Response

Reviewers Comments 

Authors Answers

Reviewer 1

Abstract section: explain the abbreviation "ROI"

ROI – Region of Interest

Swap the Discussion and Material and Methods sections in places, so that the methods are described first in the article.

Based on the manuscript author’s guidelines the order is correct.

1. Introduction

2. Results

3. Discussion

4. Materials and Methods

Unfortunately, the resolution and clarity of the figures presented are insufficient-try to improve this part of the paper-it will be of great benefit to the potential audience

The Figure 1 was change with similar image in the best resolution

Improve the References section-according to the guidelines for Authors in this journal

Reference section was improved based on the Harvard reference standard.

Reviewer 2 Report

The submitted Communication article summarizes the experience of the authors in the use of computer assisted tissue image analysis (CATIA) as optical biopsy in gynaecological cases (n=48). The authors themselves declare that “the main objective of this paper was to demonstrate the benefits of computer assisted tissue image analysis to diagnose and distinguish normal from abnormal endometrium based on texture feature analysis and machine learning and neural network classification”. In this respect, the manuscript provides a detailed description of protocols/settings in CATIA, thus resulting helpful in imaging diagnostics, but out the  main aim  of the special issue (i.e. assisting the drug discovery process and improving the clinical response to drugs).  

The manuscript lacks information on patients, the written consent for the use of medical data, and also contains 1) few typos that need correction , 2) a non-sense sentence (119 Error! Reference source not found).

Author Response

Reviewer 2

assisting the drug discovery process and improving the clinical response to drugs (main aim of the special issue)

CATIA can be a tool for diagnosis and treatment  

manuscript lacks information on patients

The Age, Gravity, number of children, Past history, etc are irrelevant for the CATIA at this stage. The addition of clinical characteristics has no added value to this paper task.

written consent for the use of medical data

Our patients participated in these studies signed a consent form that their medical records can be used for research purposes omitting their names and addresses 

few typos that need correction

See Track changes corrections

a non-sense sentence (119 Error

Reference source not found

Reference section was improved based on the Harvard reference standard.

Reviewer 3 Report

The paper seems to have been put together in a hurry from the authors' previously published work.

The organization needs to be improved,

-- Shouldn't section 4 (Materials and Methods) before Section 2 (Results) and 3 (Discussion)?

--On p.10 there is a legend for "Figure 2" while no figures are found

--On line 119, there is a message "Error! Reference source not found."

Author Response

Reviewer 3

Shouldn't section 4 (Materials and Methods) before Section 2 (Results) and 3 (Discussion)?

Based on the manuscript author’s guidelines the order is correct.

1. Introduction

2. Results

3. Discussion

4. Materials and Methods

On p.10 there is a legend for "Figure 2" while no figures

Corrected

On line 119, there is a message "Error! Reference source not found."are found

Updated

Round 2

Reviewer 2 Report

I confirm my previous report: the ms is out the aims and scope of the IJMS. I suggest submission to Diagnostics